# Collaborative Robotic Assistant Platform for Endonasal Surgery: Preliminary In-Vitro Trials

**DOI:** 10.3390/s21072320

**Published:** 2021-03-26

**Authors:** Victor F. Muñoz, Isabel Garcia-Morales, Juan Carlos Fraile-Marinero, Javier Perez-Turiel, Alvaro Muñoz-Garcia, Enrique Bauzano, Irene Rivas-Blanco, Jose María Sabater-Navarro, Eusebio de la Fuente

**Affiliations:** 1Departamento de Ingeniería de Sistemas y Automática, Universidad de Málaga, 29071 Málaga, Spain; isabelgm@uma.es (I.G.-M.); ebauzano@uma.es (E.B.); irivas@uma.es (I.R.-B.); 2Instituto de las Tecnologías Avanzadas de la Producción, Universidad de Valladolid, 47002 Valladolid, Spain; jcfraile@eii.uva.es (J.C.F.-M.); turiel@eis.uva.es (J.P.-T.); alvaro.munoz.garcia@alumnos.uva.es (A.M.-G.); efuente@eii.uva.es (E.d.l.F.); 3Departamento de Ingeniería de Sistemas y Automática, Universidad Miguel Hernández, 03202 Elche, Spain; j.sabater@umh.es

**Keywords:** force feedback, haptic teleoperation, endonasal surgery, surgical instrument navigation

## Abstract

Endonasal surgery is a minimally invasive approach for the removal of pituitary tumors (sarcomas). In this type of procedure, the surgeon has to complete the surgical maneuvers for sarcoma resection with extreme precision, as there are many vital structures in this area. Therefore, the use of robots for this type of intervention could increase the success of the intervention by providing accurate movements. Research has focused on the development of teleoperated robots to handle a surgical instrument, including the use of virtual fixtures to delimit the working area. This paper aims to go a step further with a platform that includes a teleoperated robot and an autonomous robot dedicated to secondary tasks. In this way, the aim is to reduce the surgeon’s workload so that he can concentrate on his main task. Thus, the article focuses on the description and implementation of a navigator that coordinates both robots via a force/position control. Finally, both the navigation and control scheme were validated by in-vitro tests.

## 1. Introduction

The treatment of skull base tumors may consist of surgery (open or minimally invasive techniques), radiation therapy, chemotherapy, or a combination of these [1]. Endonasal endoscopic transsphenoidal surgery (EETS) is a minimally invasive technique that is gaining wide acceptance as the first-line treatment of most pituitary adenomas [2]. There are two trends in this type of intervention, depending on the number of surgeons involved. In particular, two-handed and four-handed techniques are particularly common. In the former, a single surgeon handles the surgical instrument with one hand and the endoscope with the other, while in the latter an assistant surgeon is incorporated. Thus, in this second approach, the assistant surgeon is in charge of the endoscope and a surgical irrigator/aspirator, while the main surgeon handles two surgical instruments. Depending on the size of the adenoma and the area of the pituitary that has been invaded, one of these techniques is chosen. Specifically, for large adenomas, the four-handed technique is preferred, which reduces the incidence of residual tumour [3]. However, it complicates the execution of the intervention by requiring the coordination of two surgeons to precisely maneuver four instruments within a very small surgical space. This issue can be solved by means of the use of robotic assistant which provides accurate movements as well as coordinate its actions with the surgeon.

Despite the growth of robotic surgery in the last decade, its progress in the field of otolaryngology has been much slower. Proof of this is that as of 2013, the only system approved by the U.S. Food and Drug Administration (FDA) for head and neck surgery was the Da Vinci [4], and recently the Flex Robotic System [5] has been approved as well.

One of the main difficulties with EETS procedures for surgical robotics is the anatomical restriction of the nasal septum that limit the performance of these robotic systems. The use of the Da Vinci for EETS is inadequate due to the size of the tools, the lack of design for working with soft tissue, and the large footprint [6]. The Flex Robotic System was designed for transoral robotic surgery, which also presents limitations for the application in question [7]. Unlike minimally invasive surgery (MIS) of the abdomen, where the point of insertion in the abdominal wall or fulcrum allows some flexibility, in EETS, the efforts exerted around the nostril must be minimized because of its greater rigidity and delicacy of the tissues. So far, the work that has appeared for instrument navigation in EETS has been reduced to the study of the intranasal workspace [8].

One of the first devices designed to assist in the above-mentioned EETS technique, was the non-robotic (manual or pneumatic) endoscope holders, improving accuracy and allowing to eliminate tremors, and consequently, decreasing neurosurgeons’ fatigue. In spite of its advantages, a survey conducted by [9] suggested that the majority of the neurosurgeons preferred not to use endoscope holders in skull base surgery. The main criticism were crude movements, downward drift, loss of depth perception, lack of flexibility, cost, and bulky construction. However, the first robotic endoscope holder for neurosurgery was called Evolution 1 and was initially designed for endoscopic ventriculostomy and later modified for EETS. It is a telemanipulation robotic system with a hexapod design controlled via joystick [10]. Endoscope Robot [11] is another robotic system that has become available for clinical practice in EETS, which is made up of a positioning arm, that has seven degrees of freedom and a compact endoscope holder at its end. The Endoscope Robot is telemanipulated via a foot pedal, which consists of a joystick and different pads that ultimately control the orientation and position of the endoscope. The functionality of the Endoscope Robot was evaluated in [12], where it was shown that the interface allowed to free one of the surgeon’s hands to be able to use a second tool, improving the time of intervention, as well as its perception of the stress and exigency of the task. However, due to the limited number of participants, further studies have to be made in order to evaluate this technology.

The next technological development that expands the functionality of robotic endoscopic holders are robotic navigation systems. This means that the robot systems have teleoperated capabilities, which can include both haptic feedback and virtual fixtures guidance. The number of active developments of these systems is significantly lower compared to endoscopic holders. Three designs at an advanced stage of development will be presented below.

The first of them, the SmartArm system [13], consists of two industrial-type robotic arms with flexible tools, with a total of nine degrees of freedom each. The endoscope is held by using an external endoscope holder. SmartArm is based on a telemanipulated approach, utilizing Phantom Premium [14] haptic devices as master interfaces, governed by an impedance algorithm. The control strategy is based on their teleoperation framework in unit dual-quaternion space, which generates virtual fixtures using the vector-field-inequalities method. Virtual fixtures are generated in the slave side (the robotic arm). If a discrepancy is detected between the current and the desired position of the tool, that information is sent to the master side (haptic devices), so the user has knowledge of the direction in which the robot is finding difficulties to move. The author suggests this approach is more convenient than generating virtual fixtures on the master side and therefore affecting the slave side behavior.

In [15], another robotic system for EETS was presented. The robot is based on a parallelogram mechanism with four degrees of freedom and a contained tool adapter in order to enable the change of surgical tools during the procedure. It provides a remote center of motion (RCM) at the entrance of the nostril, improving dexterity and preventing potential damage to the body tissue. Like the SmartArm system, the robot is telemanipulated, but instead of using haptic devices, a joystick is utilized. For this reason, although virtual fixtures are implemented, it is not possible to use the approach described in [13], and constraints will be generated in the side of the master. The main drawback is the lack of haptic feedback, depending only on visual feedback.

Finally, [16] proposed a robotic surgical system based on the Smart Arm concept, composed of two industrial robot arms with six degrees of freedom. Its focus is the development of a more intuitive and dexterous interface that can cover the entirety of the EETS phases. The proposed system follows a hybrid approach, where a force-controlled interface and a serial-link interface are attached to each robot arm, holding a multi degree of freedom forceps at its end. The neurosurgeon exerts force over a vertical handle attached to a six-axis force/toque sensor, which ultimately, via an admittance control algorithm, operates the robot arm. However, the serial-link interface, with six degree of freedom, decouples the movements of the robotic arm from the action taken by the neurosurgeon. This hybrid approach avoids the mirror effect while allowing to implement control techniques such as virtual fixtures and virtual remote center of motion.

As a conclusion, the robotic systems described in this background explore the use of teleoperated robots to assist the main surgeon. This work aims to take a further step towards a four-handed endoscopic endonasal surgery by means of a robotic system equipped with a remote-operated robot and an autonomous one. Specifically, the main contribution of this work is the development of a framework for a robotic surgical platform that allows the collaboration of a robotic arm teleoperated by the surgeon and an autonomous robotic arm in endonasal surgery procedures. This platform is based on robotic operating system (ROS) and Matlab for testbed proposes. In this way, a navigation system for human–robot interaction was developed, which plans the movement references for the force/position control of each manipulator of the robotic platform.

The paper is organized as follows. The next section presents the problem statement, as well as the functional architecture proposed in order to explore the collaboration between the surgeon and the robotic platform. Section 3 describes the scenario designed to validate the proposed architecture, the experiments performed, and the results obtained. Finally, Section 4 is devoted to discussing the results obtained, summarizing the contributions of the paper, and proposing future works of this research.

## 2. Materials and Methods

The surgical procedure considered in this work is an endonasal surgery to remove a pituitary tumour (EETS). In this procedure, the objective is the sarcoma resection approaching the operation through the nostrils. The protocol for this procedure consists firstly of nasal widening, then instruments are inserted through the nostrils and various dislocations of the nasal turbinates are performed, creating a pathway to the sphenoid. Once this area is reached, the posterior wall of the sphenoid is dissected and sometimes the tumour will already be present as a result of a rupture of the sella. If this is not the case, a new incision will be necessary to finally remove the tumour by means of traction and avoiding excessive force that could damage the surrounding structures. In particular, care must be taken in this operation to avoid affecting the carotid artery and the optic nerve.

The reduced protocol considered in this paper to explore human–robot interaction will focus on the last stages of this procedure: the opening of the dura mater and the removal of the tumour. The main objective is to propose a robotic surgical scenario for endonasal surgery, which decreases the surgeon’s workload, while at the same time providing more precision in the surgeon’s movements. In this way, the surgeon will be considered to teleoperate, with haptic feedback, a robotic arm that manipulates the anatomical structures, while a second arm will autonomously assist in cutting the dura mater. Figure 1 shows on the left the movement of the instruments within the nasal septum during this intervention and on the right the proposed robotic system together with the set of reference frames used. Thus, in a) the location of the global frame {W} in the columnello-septal complex is included. The movement of the surgical instrument in order to carry out a displacement, in this case from A to B, must pivot on the point ${}^{W}F_{P}$ referred to {W}. Figure 1b shows complete proposed robotic assistant compose of a teleoperated arm, an autonomous arm, a haptic device, and the surgical area. The coordinate frames that will be used to define the navigation task of the surgical instruments carried by the two robots were indicated on each component of the robotic assistant. Therefore, both the positions of the surgical instruments and the points of interest introduced in the pre-operative phase are referred to the global reference frame {W}. On the other hand, the systems associated with the robot base, the end effector, and the distal end of the surgical instrument are named {R1}, {E1}, and {TCP1} for the teleoperated robotic arm and {R2}, {E2}, and {TCP2} for the autonomous robotic arm. Finally, the reference frame {C} was associated with the haptic device that interfaces with the teleoperated robot. For simplicity, in the scheme, it was only considered one ${}^{W}F_{P}$, but in the actual system, each surgical instrument needs its own ${}^{W}F_{P}$.

The implementation of the functional architecture, for the proposed robotic assistant, by using ROS (robot operating system) is detailed at Figure 2. Thus, the blocks colored in green correspond to the ROS nodes that configure the components of the functional architecture. On the other hand, the ROS topics, through a publisher–subscriber relationship, define the connections between the different components. In this way, OMNI node is in charge of the management of the haptic device and NAV-MATLAB node contains the high-level functions that are implemented in Matlab. These functions include the human–robot interaction and the motion control planning of the manipulators. In this way, NAV-MATLAB sends the position references to the force/position control of each manipulator, represented by CRANEEAL-1 and CRANEEAL-2 nodes. These last nodes send the velocity reference to the UR-1 and UR-2 nodes in charge of managing the teleoperated arm and the autonomous arm, respectively. These two nodes also provide the end-effector position of the manipulators, which combined with the interaction forces provided by HEX-1 and HEX-2 (force/torque sensors) close the force/position control loop. Finally, two additional ROS nodes are considered to handle all the transformations between the coordinate systems detailed in Figure 1b (TF node), and CAMERA node that provides the endoscopic image.

The following subsections will detail the Navigation and Human–Robot Interaction modules corresponding to NAV-MATLAB node as well as the Force/Position control module included in CRANEEAL-1 and CRANEEAL-2 nodes.

### 2.1. Human–Robot Interaction and Surgical Instruments Navigation

The surgeon interacts with the robotic assistant through a haptic device that sends the incremental position reference for the distal end of the instrument carried by the teleoperated robot {TCP1} to the Navigation module. Moreover, the haptic device provides the surgeon with the feedback forces, both instrument interaction and virtual forces that are used for guiding {TCP1} towards the surgical target. On the other hand, in the preoperative phase, the surgeon introduces into the Human–Robot Interaction module the following information:the required sequence of human–robot collaboration that will carry out in the reduced surgical protocol,the situation of the carotid artery and the optic nerve, which are considered the obstacles,a cylindrical virtual fixture that configures the nasal septum working area,the situation of the dura mater and the tumour.

The Human–Robot Interaction module uses this information to define the state machine that schedule the movements of the two robotic arms in such a way that each state represents a cooperative human–robot surgical maneuver. This means that every state includes information for planning the movements and behaviors of both robot arms. In particular, the data related to the autonomous robot are the type of surgical instrument that will carry during the maneuver, the goal position to reach inside the nasal septum and the goal force that has to be exert. If the surgical instrument is an endoscope, this goal force will be null, but this parameter has to be defined for other surgical tools like a drilling device. On the other hand, for the teleoperated robot, a goal position is also included, used for defining the straight ideal trajectory from the current {TCP1} location. This ideal trajectory will be employed to guide the surgeon, during the haptic teleoperation, in order to reach the desired goal position. If this parameter contains a null value, no ideal trajectory is considered during the current surgical maneuver. In this way, the state machine, included in the Human–Robot Interaction module, sends both the goals position and force to the autonomous robot, enables the haptic device, and defines the ideal trajectory (if considered) to operate the teleoperated robot. The transition to the next state of the surgical protocol is defined either when the autonomous robot reaches the goal position with the commanded goal force or by a surgeon order.

The Navigation module is based on a potential field navigation strategy for the movement command generation for the two robotic arms. Thus, the Navigation module is composed of three subsystems (indicated in Figure 3 by three rectangles colored in green) and it generates the position/force references for the teleoperated and autonomous robots by taking into account the repulsion forces due to the considered obstacles.

The References for teleoperated robot subsystem (upper green rectangle at Figure 3) receives from OMNI node the position reference of the haptic device related to its own reference frame {C}. This reference is transformed to the global working frame {W} by means of the ${}^{W}T_{C}$ transformation. This transformed reference is scaled through the gain $K_{sp}$ and is sent to the force/position control subsystem of CRANEEAL-1 node. The interaction forces provided by the force sensor of HEX-1 node are used for closing the haptic control loop of the teleoperated instrument. A first order holder is used to adequate the data reading frequency of the force sensor to the operating frequency of the haptic device. These held interaction forces are scaled by $K_{sf}$ and added to the repulsion forces generated by the Repulsion forces for obstacle avoidance subsystem as well as the attraction forces for reaching the Teleoperated Robot’s goal position, which is computed by scaling the distance to the ideal trajectory. Finally, the resulting forces, transformed by means of ${}^{C}T_{W}$ to the reference system {C}, are used by OMNI node for providing the surgeon with haptic sensation.

The repulsion forces $F_{re1}$ (subsystem Repulsion forces for obstacle avoidance) are composed of the virtual fixtures that limit the surgical area ($F_{v}$), the repulsion forces of the obstacles ($F_{o}$) and the repulsion forces generated by the tool carried by the autonomous robot ($F_{a}$). Each of these repulsion forces are generated by means of a potential function [17] that uses the geometric models of each of the aforementioned elements and the current positions of the two robots. Moreover, this sub-system also generates the repulsive forces for the autonomous robot, $F_{re2}$, which is computing in the same way as $F_{re1}$, but substituting $F_{a}$ by the repulsive force exerted by the instrument of the teleoperated robot $F_{t}$.

Finally, the References for autonomous robot subsystem calculates a reference force by adding the following four elements: repulsive forces for autonomous robot, interaction forces HEX-2 provided by HEX-2 node, goal position attraction force, and scaled goal force. In this way, goal position attraction force is calculated by scaling the distance to the goal of the surgical instrument by the stiffness $K_{a}$. On the other hand, the scaled goal force is the target force provided by the Human-Robot Interaction module scaled by $K_{g}$. Finally, the reference force is converted to an incremental position reference by considering a unit virtual mass and integrating it twice. This reference is sent to the Force/Position control managed by CRANEEAL-2 node.

### 2.2. Force Feedback Control System for the Autonomous Robotic Co-Worker

The use of robots in Endonasal Transsphenoidal Surgery requires adaptation of the control strategy to the intervention to be performed. In the removal of tumors at the base of the skull, the different stages of surgery require different robot behavior. We apply a force feedback control strategy to control its movements:When the robot moves in areas close to the carotid artery or optic nerve, to avoid or minimize contact.When the dura mater is removed, as the movements must be very precise.During tumour removal, as more force must be exerted, but always below a specified threshold.

The autonomous robotic co-worker is a UR3, a collaborative robot from Universal Robots^®^. The trajectory of the autonomous robot is planned from the current position using OMPL (Open Motion Planning Library), which is the MoveIt default planner. The time that elapses from the moment in which the target point is determined and the moment in which the robot begins to move is around 1 s, depending on the complexity of the movement. A control loop with a low frequency would lead to unstable behavior of the robot, since it would not have time to react to the interaction forces that might appear. Furthermore, the endonasal cavity is a small working space: it can be approximated by a cylinder with a diameter of 27 mm and a length of 70 mm, as indicated in [8]. The alternative is to send URScript commands through sockets to the robot controller. It is the most direct way to control this, and therefore, the one that guarantees the updating of setpoints at the highest supported speed (125 Hz).

Various commands associated with movement can be found in the documentation provided by Universal Robots on URScript [18], the most relevant of which being MoveJ, MoveL, ServoJ, SpeedJ and SpeedL. To determine which movement command is the most suitable, an experiment was carried out by bringing the robot to an initial position that would allow it to move freely 100 mm in the positive direction of the X axis. Orders were sent using the different commands available so that it reached the final position of 100 mm at a constant speed of 10 mm/s. The command was updated every 8 ms (125 Hz). SpeedJ and SpeedL were the commands that worked best for real-time control. The movements were smooth and predictable. Although they have the disadvantage that the positioning of the robot is more complicated, since a speed is commanded directly instead of a position, these have been the commands used to control the movement of the autonomous robot with the control algorithm with force feedback.

The references generated by the Navigation module in order to reach the goal position define the trajectory of the surgical tool carried by the autonomous robot. A force feedback control system has been developed, which will be active while the robot carries the surgical tool, minimizing its contact with the internal structures of the endonasal cavity. It is intended to have a repulsive behavior, so that when it detects force in a certain direction and sense, it moves in the same direction, but in the opposite sense, with the aim of reducing the value of that force. It is important to mention that, because of the holonomic constraints it will only act on the position of the surgical tool, and the orientation a function of the current position and the pivot point ${}^{W}F_{P}$. Figure 4 shows the diagram of the force feedback control system that has been implemented.

The interaction between the robot and the endonasal cavity generates the Interaction forces that are measured by the force sensor (HEX-1 for teleoperated robot and HEX-2 for autonomous robot). This signal is filtered using a moving average filter (which will be described later). Next, a force-position transformation is performed using the $K_{F}$ constant (mm/N) (whose value will be determined later). Thus, the Position compensation due to interaction forces is obtained, which constitutes the feedback of the system. The position and orientation of the tool center point (TCP) (HOPKINS endoscope), is used to calculate the Tip Tool Position, taking into account the ${}^{W}F_{P}$ (Fulcrum point). Blocks ${}^{i}T_{j}$ represent the transformation of positions and forces from the reference frame j to the reference frame i.

As the block diagram shows, from Interaction forces and End Effector Position the Desired End Effector Position is calculated, which once transformed (${}^{R}F_{W}$) to coordinates of the reference system associated with the base of the autonomous robot, is defined as the input to the Proportional-Derivative (PD) controller. The output signal of this controller is the position control input, which will be transformed into a velocity reference, which constitutes the input to the robot controller, using the SpeedJ and SpeedL commands, previously indicated.

The Ziegler-Nichols method has been used to tune the PD. The robot TCP has been subjected to a 10 mm step-type command for 2 s, after which it returns to a null value. The acceleration value has been fixed at 80 mm/s^2^. Due to the small dimensions of the nasal cavity, the objective is to guarantee the stability of the robotic arm inside it for step-type profiles of the command signal. The values obtained for the proportional and derivative gains of the PD controller have been $K_{P}=0.13$ and $K_{D}=0.032$.

The design of the filter stage of the interaction force signal with the endonasal cavity, measured with the force sensor, has been carried out with Matlab and Simulink software. First, a simulation was performed by applying a low pass filter to the force signal for the cutoff frequencies: 20 Hz, 10 Hz, 5 Hz y 3 Hz. Filtering is more effective as the cutoff frequency decreases, although at the cost of increasing the delay. Second, the moving average filter was applied to the force signal, using window sizes values of 16, 32, 64 and 128. The results obtained with both simulations show that the moving average filter with a window size of 32 is the one that provides the lowest noise level and the least delay, as shown in Figure 5.

## 3. Experiments and Results

In Figure 6a the scenario designed to validate the proposed functional architecture is presented. The four main elements are: the teleoperated robotic arm, the autonomous robotic arm, the haptic interface device and the 3D model of the human skull made by 3D printing. The reference systems indicated in Figure 1 have been defined on these four elements. Figure 6b shows the reference system {W} defined on the skull model. This is the common reference system in the scenario when the two robotic arms are used. For reference, the dimensions of the cube that encompasses the entire volume of the skull model are 142 × 198 × 91 mm.

The autonomous robotic arm is a collaborative robot, a UR3 model from Universal Robots^®^. It has a repeatability of ±0.1 mm. The internal control loop operates at 125 Hz, which limits the maximum frequency of the control system to be developed.

The control system calculates the force exerted on its TCP from the current consumed by the robot’s servo motors. This results in the sensitivity in the estimation of force, in the best of cases, being around 5 N [18], which is clearly insufficient for this application of endonasal surgery. In [19] results are provided showing that average forces recorded during in vivo testing skull base surgery ranged from 0.1 to 0.5 N, and the average maximal force was 1.61 N. This has led us to incorporate a force sensor, the HEX-E model from OnRobot^®^, on the autonomous robotic arm. It is a 6-axis sensor that allows the measurement of forces and moments in the XYZ axes. Its nominal capacity is 200 N and 10 Nm and its maximum sampling frequency is 500 Hz.

The experimental scenario has been completed using surgical material from Karl Storz^®^ [20] for Endonasal Trans-sphenoidal Surgery:Model 28132 BA endoscope with HOPKINS^®^ optics, with a visual direction of 30°, an outer diameter of 4 mm and a length of 18 cm.Model 649179 B vacuum cleaner with flexible tube, conical tip, 18 cm long and 1.35 mm outer diameter.Model 28164 ED Ball end coagulation electrode, with a diameter of 2 mm and a total length of 13 cm.Model 252682 drill. It has a rotational speed of 100,000 rpm, a useful length of 93 mm, an outside diameter of 7.5 mm. It accepts milling cutters with a shaft diameter of 3.17 mm.Model 28164 KB curette with spoon shape, angled end, a size of 2 mm, round handle and length of 25 cm.

An Ubuntu 16.04.6 LTS distribution has been used in which the Kinetic Kame version of ROS (Robot Operating System) has been installed. The programming has been done in C++. The MoveIt Motion Planning Framework is the most widely used work environment dedicated to robotic manipulation and motion planning in ROS, being compatible with our UR3 robotic arms. The Move Group Interface has been used for basic functionality and MoveIt ROS for more advanced movements. The tf2 ROS package has been used, which allows managing multiple coordinate systems. The 3D viewer Rviz has also been used, which allows obtaining the three-dimensional model of the robot, seeing its movements in real time and visualizing the reference systems and the relationships between them.

The following subsections present the experiments carried out on the robotic platform in order to validate the proposed control architecture. Firstly, the validation of the Force/Position control system is detailed, with two scenarios designed for the performance of the autonomous robot. Next, the trials to validate the haptic system are described, focusing on the study of the attraction and repulsion forces generated to guide the surgeon towards the goal point.

### 3.1. Force/Position Control System Validation

In order to validate the control system in an endonasal surgery scenario (see Figure 6), different collision and contact scenarios were proposed, which mimic possible situations that may appear during an endonasal surgery intervention. To configure these scenarios, 3D printing was carried out, using polylactic acid (PLA) (Figure 7a). The protruding element that can be seen in this piece allows it to be anchored to the skull model (see Figure 7b). This piece has a cylindrical cavity 52 mm in length and 12 mm in diameter, the objective of which is to create a scene as similar as possible to a nasal cavity.


**Scenario 1:**


The geometry of this scenario is shown in Figure 8a. The tool carried by the robot was an endoscope, and the desired trajectory was a linear one parallel to the Z axis (shown with a dashed white line), to force contact with the wall. The starting point of this trajectory was located at (X,Y,Z)=(−4.5, 0, 0) mm and the final point was located at (X,Y,Z)=(−4.5, 0, 50) mm. The objective of this test was to analyze the behavior of the robot with the developed force feedback control strategy, when the endoscope comes into contact with the side wall of the scenario.

To carry out the tests, the parameters shown in Table 1 were considered. The sensor used has a “force threshold”, which is the measurement threshold below which it returns a null value.

In Figure 8b, the force measured and filtered (F_X_-blue color), and the position (TCP_X_-red solid line) on the X axis of the robot TCP are shown. The desired path of the TCP is shown as a red broken line (TCP_X_ SP–set point), characterized by X=−4.5 mm. As time increased, the robot’s TCP deviated from the desired path (X=−4.5 mm), and reached approximately X=−2 mm, as can be seen in Figure 8b. This behavior was desirable, since it indicated that the endoscope was moving to the left, due to colliding with the scenario wall (a ramp). The force measured on the X axis increased as the robot deviated more and more from its desired path.

Figure 8b shows a “stepped” evolution of both the force on the X axis ($F_{X}$) and the position of the TCP (TCP_X_). This was because when the tool collided with the channel wall, it moved in the direction normal to the contact surface, a distance determined by the value of $K_{F}$, after which the robot tries again to follow the reference, and therefore (due to the scenario geometry), it detects an increase in the contact force with the wall and tries again to compensate it.

Figure 8c shows the measured and filtered force ($F_{z}$-blue color) and the position (TCP_Z_-red solid line) on the Z axis of the robot’s TCP. In a dashed red line (TCP_Z_ SP), the desired path of the TCP, on the Z axis, is shown, which evolved from Z=0 mm to Z=50 mm.

The force measured in the Z axis (Figure 8c) was significantly less than the force measured in the X axis due to the steepness of the stage ramp. Therefore, in the interval [0, 11] seconds, the magnitude of the force on the Z axis remained at values very close to zero. From that moment on, the force measured on the Z axis increased because the robot deviated each time from its desired path.

Figure 8d shows in 3D the desired and the actual trajectories of the robot throughout the experiment.


**Scenario 2:**


Scenario 1 is printed on stiff material, and does not absorb any strain. To simulate an environment “closer” to the nasal cavity, a new scenario was created using EVA rubber to simulate the internal tissues of that cavity. Two layers of 3.8 mm each were used. A schematic of this second scenario is shown in Figure 9.

The path to be carried out by the robot evolved from Z=0 mm to Z=20 mm. When the robot reaches Z=2.4 mm, the TCP will come into contact with the EVA rubber. At Z=10 mm the TCP will collide with the worktable and, therefore, will not be able to advance any further. Figure 10a shows the measured and filtered force ($F_{z}$-blue color) and the position (TCP_Z_-red solid line) on the Z axis of the robot’s TCP. A red dashed line (TCP_Z_ SP) shows the desired path of the TCP.

In Figure 10a, we can distinguish three zones:
0 s<t<1 s: The endoscope has not come into contact with the EVA rubber, and therefore the forces that appear are noise and the position is not greatly affected.1 s<t<4.5 s: Elastic behavior is observed in this interval. As the robot TCP advances, the forces increase proportionally and stably, since no oscillations are observed.4.5 s<t<7.5 s: The TCP of the robot has reached the table. An oscillatory behaviour of the force is observed, of increasing amplitude. At time t=7.5 s, the force reaches the maximum value, and the execution of the experiment is stopped.

Therefore, the results shown in Figure 10a are not adequate: the measured force and the position of the robot’s TCP show unacceptable oscillations. To eliminate these oscillations, a set of experiments were carried out in which: the constant $K_{F}$ was varied between 1 mm/N and 5 mm/N, the size of the filtering window was increased from 32 to 64, and the force threshold varied between 0.1 and 0.3 N.

Figure 10b shows the results obtained for $K_{F}=3\;\mathrm{mm}/N$, a filter window of 64, and a force threshold of 0.2 N. The measured and unfiltered force (violet color) and the measured and filtered force (blue color) on Z axis are shown. As can be seen, the filtering of the force signal was quite effective, and it was also capable of absorbing large jumps in the signal, notable from the instant t=5 s, providing more stability to the system.

### 3.2. Haptic Teleoperation Trial

The proposed experiment shows the haptic navigation of the teleoperated instrument, which is commanded by the surgeon by using the haptic device. Therefore, this device sends to the Navigation module the successive incremental positions for {TCP1} related to the {C} reference frame. In this way, this module computes the locations of the mentioned system with respect to {W} by pivoting around ${}^{W}F_{P}$. Figure 11 presents the trial scenario where a first cylinder is a virtual fixture to delimit the working space inside the nasal septum. This working space was modelled by a truncated cone with 17.5 mm mayor radius, 9.4 minor radius, and 77.4 mm height. On the other hand, the surgical instrument carried by the autonomous robot was represented by a cylindrical shape of radius 3.2 mm. Due to the use of standard laparoscopic instruments in this trial instead of endonasal surgery instruments, which were smaller than laparoscopic ones, the proposed endonasal scenario was scaled by 1.2. Both cylinder and truncated cone generated repulsive forces on the teleoperated tool that were fed back to the surgeon via the haptic device. Finally, the surgeon defined in pre-op stage the position of the sarcoma center at (X,Y,Z)=(−1.9, −8.0, 95.4) mm related to {W}. This point defined a line that connects this point to ${}^{W}F_{P}$, which produces attraction forces for guiding the surgeon to the defined target. In this way, the teleoperated trajectory commanded by the surgeon is shown as a stream of red points, which are contained inside the working area and avoid the other surgical tool.

Figure 12 displays in detail both the teleoperated trajectory (a) and the generated forces during this trial (b). In this trial of 45 s, the surgeon tried to collide with autonomous surgical instruments, but the repulsion forces generated by the cylinder that models this tool avoided this situation and the teleoperated tool surrounded the obstacle without bumping. Figure 12a presents the teleoperated trajectory (the stream of {TCP1} related to {W} positions) with the cylinder that models the surgical tools carried out by the autonomous robot and the line that connected ${}^{W}F_{P}$ with the target point, named reference trajectory.

This teleoperated trajectory produced virtual forces, which are plotted at graph 12b. These forces were related to the {C} reference frame attached to the haptic device and provided the surgeon with haptic feedback. It was observed that the repulsion forces increased when the surgeon tried to touch the autonomous robot surgical tools during the 5 s to 32 s of the teleoperated trajectory. This had the effect that the teleoperated trajectory surrounded the obstacle, as was detailed before. On the other hand, when the surgeon followed the reference trajectory, the resulting virtual forces were almost null, with small damping. In this way, both robots, teleoperated and autonomous, can share safely the surgical area avoiding accidental collisions.

## 4. Discussions and Conclusions

The endonasal surgery for sarcoma resection offers new trends to apply robotics for minimally invasive surgery. In particular, this kind of approach needs special skills and accuracy during the surgical maneuvers, since at least two instruments have to share a tiny surgical area where there are vital structures that must not be damaged. The technological response to this challenge first starts with camera holder devices, which evolves to teleoperated robot arms combined with the virtual fixtures concept. Depending on the interface type, robotic endoscope holders can be divided into two categories: telemanipulation or cooperative mode. Telemanipulation mode takes an input from an external source, like a joystick, haptic device, etc., and, as a result, the robot moves accordingly. Instead, cooperative mode enables the neurosurgeon to operate the robotic system directly, making it more intuitive and convenient [21]. In this way, such robotics solutions provide with accuracy and safety to the surgeon during the surgical procedure.

None of these solutions incorporates mechanisms to implement collaborative tasks in neurosurgical interventions. Nevertheless, in the field of abdominal surgery these techniques were implemented with very promising results. Thus, we can find collaborative robotic assistants that perform pre-planned automatic tasks in certain surgical maneuvers depending on the current state of the operation [22], or assistance based on automatic navigation of the endoscope using learning by demonstration techniques [23]. In order to reproduce human movements, regression techniques [24] and the modelling of robot dynamics [25] were also used. To increase the collaborative skills of the systems, some works implemented cognitive architectures that provided assistants with greater decision-making capacity and incorporated mechanisms to improve the behavior of the robot based on corrections made by medical personnel [26].

This paper proposes a new concept based on the combination of a teleoperated robot arms with an autonomous one. The aim of this work was focused on the design of an open testbed robotic platform able to validate minimally invasive surgery human–robot interaction. For this propose, a functional architecture able to coordinate the human actions with the two robot arms via a two-level control motion strategy was developed. The high-level motion control was composed of a Human Robot Interaction block and a Navigation module, which managed the coordination of the two robotics arms and provided the surgeon with haptic feedback computed with both real interaction forces and virtual ones. This feature allowed a simultaneous safe movement of both robot arms, since the robot’s motion took into account the virtual fixtures, which enclose the surgical area as well as avoid other obstacles like vital structures or other surgical tools, which share the surgical area. The second control level was materialized on a Force/position control level. This module covered the incremental positions sent by the haptic device on velocity references, which commanded the robot actuators in order to displace the surgical tool. It used a proportional derivative strategy followed by a classical cartesian controller. The proposed force/position controller sent feedback regarding interaction forces and it considered the holonomic motion constraints because of the natural insertion point to the nasal septum. This feature avoided unexpected damage to the columnello-septal complex due to inappropriate movements.

The proposed robotic platform for endonasal surgery will allow the development of research lines related with to human–robot interaction. The next step will be to focus on exploring the human–robot interrelationship during a simplified protocol for the extraction of a sarcoma. This collaboration will be based on the information obtained in the preoperative phase, which included both the three-dimensional model of the patient’s skull and the identification by the neurosurgeon of the anatomical areas of interest. In addition, the robotic system will allow off-line planning of surgical actions through simulation and modelling of the intervention. In particular, the use of ontologies for representing the surgical procedure knowledge will provide the robotic system with new skills for taking decisions during the surgical procedure as well as on-line learning. This skill will need a second research line regarding a perception system able to recognize the current step of the surgical protocol that the surgeon is carrying out, as well as unexpected situations. These lines are designed for making a step forward from a teleoperated robot system to a collaborative robot system in surgery.

## Figures and Tables

**Figure 1 sensors-21-02320-f001:**
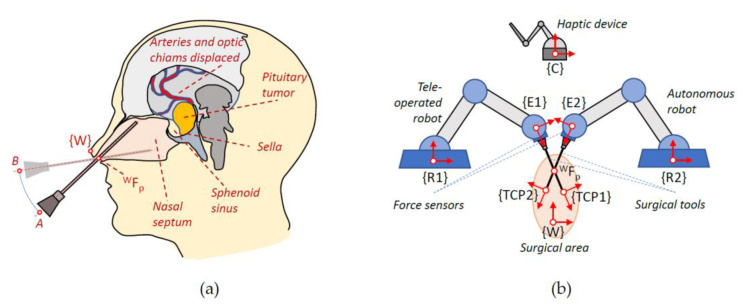
Surgical scenery. (**a**) Movement of the instruments inside the nasal septum. (**b**) Robotic assistant and reference frames.

**Figure 2 sensors-21-02320-f002:**
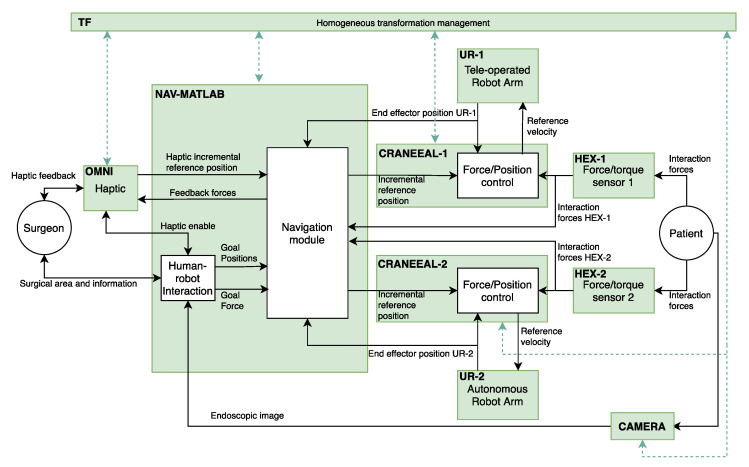
Robotic operating system (ROS) implementation of the Functional Architecture.

**Figure 3 sensors-21-02320-f003:**
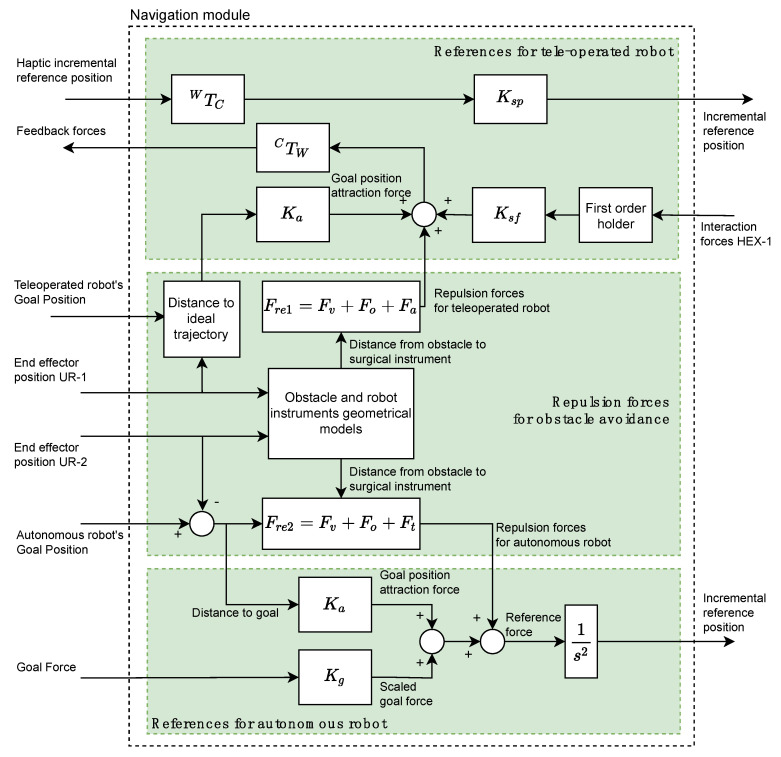
Navigation module and subsystems.

**Figure 4 sensors-21-02320-f004:**
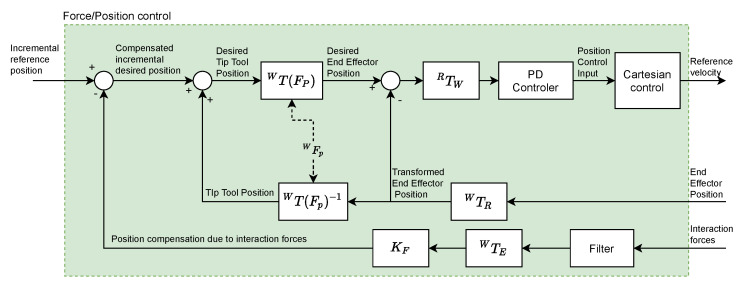
Block diagram of the control system with force feedback.

**Figure 5 sensors-21-02320-f005:**
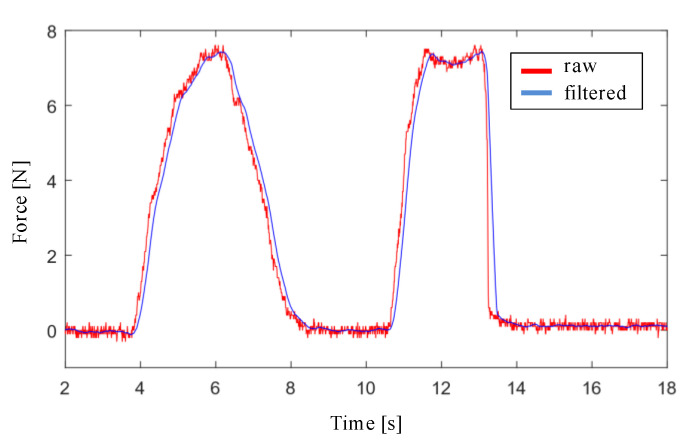
Result of filtering the force signal with a moving average filter with a window size of 32.

**Figure 6 sensors-21-02320-f006:**
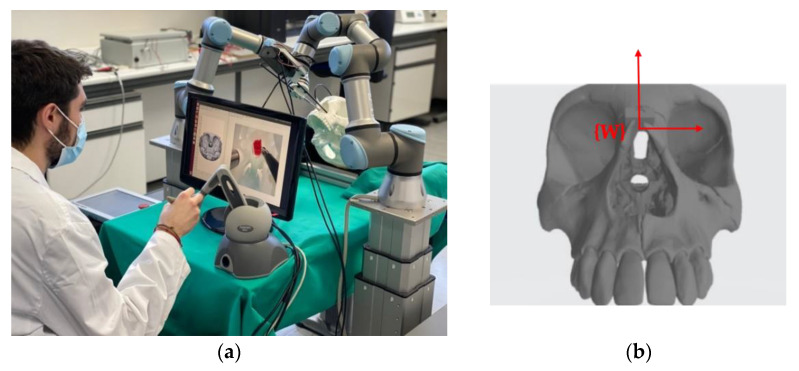
(**a**) Endonasal surgery setup. (**b**) Reference system in the 3D model of the skull.

**Figure 7 sensors-21-02320-f007:**
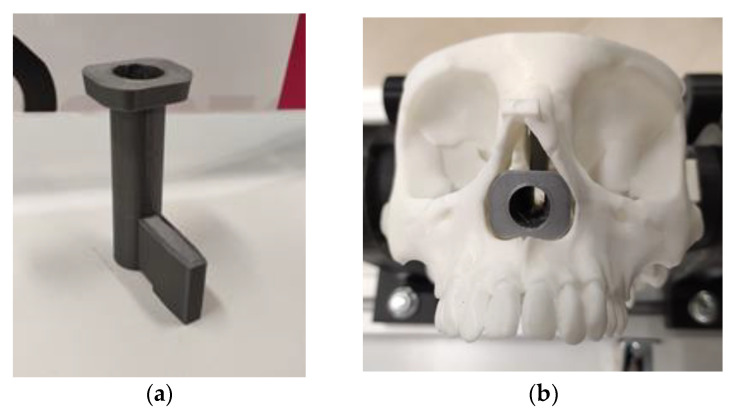
Piece manufactured to characterize the interior of the nasal cavity. (**a**) Piece made with 3D printing. (**b**) Piece placed on the skull model.

**Figure 8 sensors-21-02320-f008:**
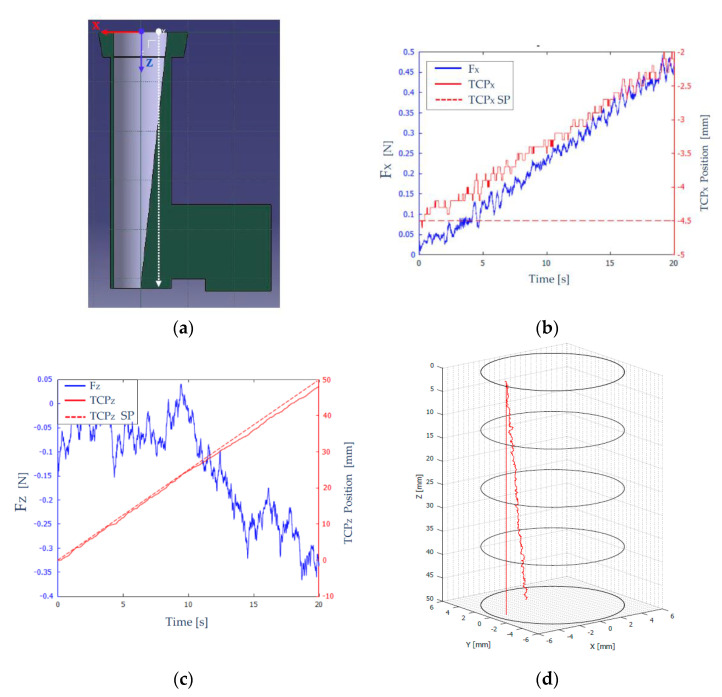
Results obtained in scenario 1. (**a**) Scenario 1 section. (**b**) Forces (blue) and position (red) on the X axis of the robot’s TCP. (**c**) Forces (blue) and position (red) on the Z axis of the robot’s TCP. (**d**) 3D representation of the robot’s TCP position.

**Figure 9 sensors-21-02320-f009:**
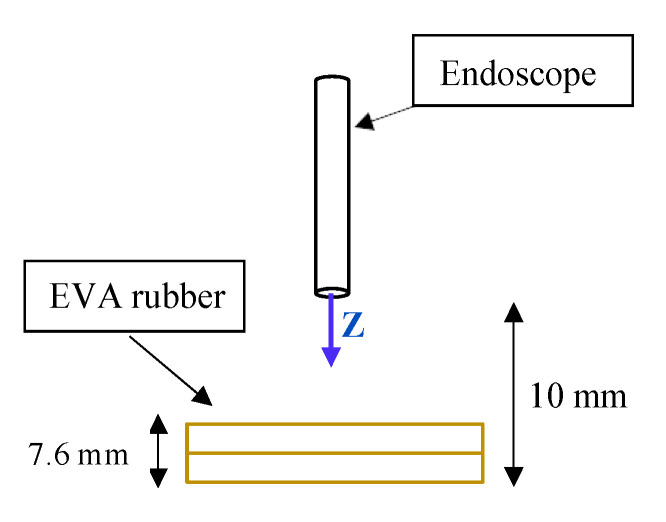
Scenario 2 schematic.

**Figure 10 sensors-21-02320-f010:**
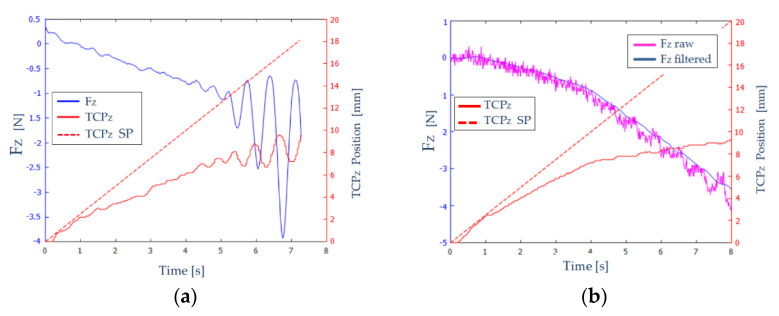
Results obtained in scenario 2. (**a**) Forces (blue) and position (red) on the Z axis of the robot’s TCP for $K_{F}=5\;\mathrm{mm}/N$ (**b**) Forces (unfiltered force–violet line, filtered force–blue line) and position (red) on the Z axis of the robot’s TCP for $K_{F}=3\;\mathrm{mm}/N$.

**Figure 11 sensors-21-02320-f011:**
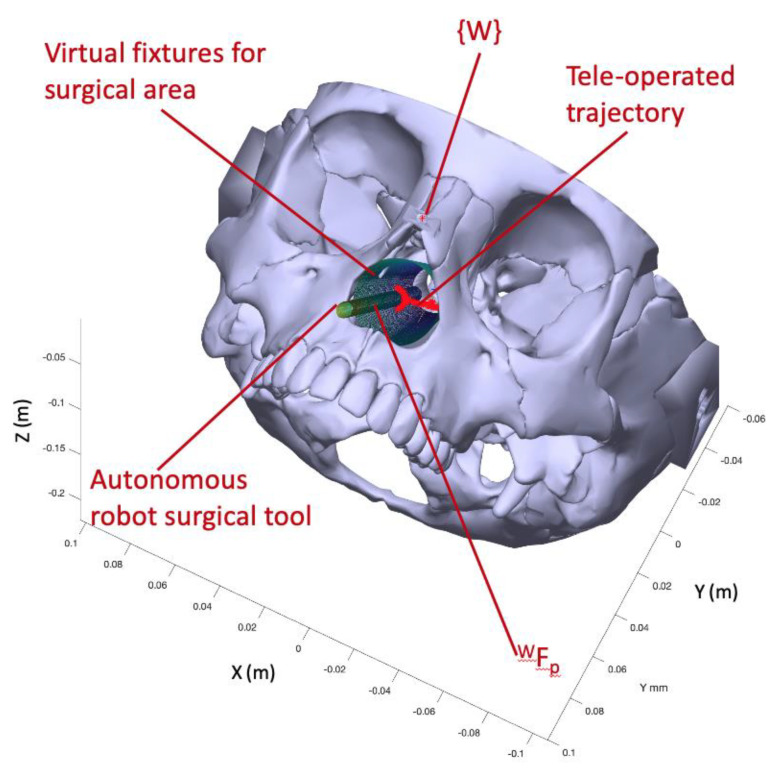
Proposed scenario for teleoperated navigation test.

**Figure 12 sensors-21-02320-f012:**
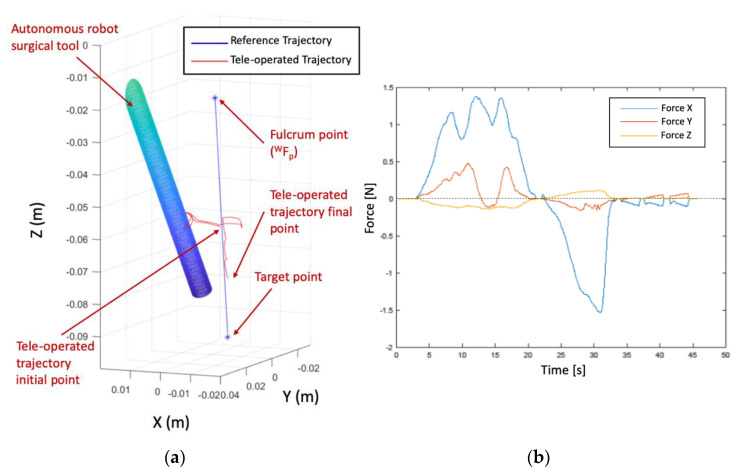
(**a**) Teleoperated trajectory and (**b**) computed virtual forces during the trial.

**Table 1 sensors-21-02320-t001:** Parameters used in the test scenario.

Proportional-Derivative PD controller gains:	$K_{P}=0.13$, $K_{D}=0.032$
Force threshold:	0.1 N
$K_{F}$:	5 mm/N
Robot’s tool center point (TCP) speed:	2.5 mm/s
Robot’s TCP acceleration:	80 mm/s^2^

## Data Availability

Not applicable.

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
