# Peer review of "Collaborative Robotic Assistant Platform for Endonasal Surgery: Preliminary In-Vitro Trials"

_sensors, 2021, doi:10.3390/s21072320_

Round 1

Reviewer 1 Report

  1. Briefly summarize the content of the manuscript;

Very nice study. I enjoyed reading it. The experiments and the results take a step further in medical robotics, lack of haptic sense and/ or autonomy of the current robotic system are drawbacks for the current robotic medical systems.  The authors proposes a concept combining a tele-operated robot arm with an autonomous one for endonasal pituitary tumor surgery. Both autonomous and tele-operated robotic arms provide haptic control systems.

  1. Illustrate what are, in your opinion, the manuscript’s strengths and weaknesses;

+ The presented concept has a high level of novelty and originality, the experiments are well presented and in a logical manner. There is sufficient information. 

- Even though in do not consider it as a major impediment for the publication, there is a kind of imbalance between different sections of the manuscript that makes it less easy to follow.  There is an abundance of information (characteristics and limitations of the previous robotic systems, Line 47-138) in the introduction section (more than 3 pages). However, the discussion section is less comprehensive and it limits to the summarization of the current concept. As it is, some of the readers may have to return to the introduction section in order to understand and integrate the results of the study. Perhaps, it may be more suitable to use some of this information in the discussion section. They can better serve as a reference/ comment point for the results of the current study.

  1. Provide a point-by-point list of your major recommendations for the improvement of the manuscript;

I have no major recommendations

  1. If necessary, provide a point-by-point list of your minor recommendations for the improvement of the manuscript.

Line 45 – FDA; Abbreviations should be defined in parentheses the first time they appear

Line 49 – The statement “The use of the Da Vinci for EETS is inadequate due to the size of the tools, the lack of design for working with soft tissue and the large footprint” requires proper citation

Line 341 – PD; Abbreviations should be defined in parentheses the first time they appear

Line 334 – TCP; Abbreviations should be defined in parentheses the first time they appear

Line 378 – The statement: “which is clearly insufficient for this application of endonasal surgery” is insufficiently substantiated.

Line 416 - PLA; Abbreviations should be defined in parentheses the first time they appear

Author Response

The reply is included in the author cover letter

Reviewer 2 Report

Please see the attached review comments.

Author Response

(The authors gave the same response as above.)
